# Technical Note: An efficient method for accelerating the spin-up process for process-based biogeochemistry models

Yang Qu[1], Shamil Maksyutov[2], and Qianlai Zhuang[1,3]

[1]Department of Earth, Atmospheric, and Planetary Sciences, Department of Agronomy, Purdue

University, West Lafayette, IN 47907 USA

[2]National Institute for Environmental Studies, 16-2 Onogawa, Tsukuba, Ibaraki, 305-8506 Japan

[3]Department of Agronomy, Purdue University

To be submitted to: Biogeoscience

Correspondence to: qzhuang@purdue.edu

**Abstract**
To better understand the role of terrestrial ecosystems in the global carbon cycle and their
feedbacks to the global climate system, process-based biogeochemistry models need to be
improved with respect to model parameterization and model structure. To achieve these
improvements, the spin-up time for those differential equation-based models needs to be
shortened. Here, an algorithm for a fast spin-up was developed by finding the exact solution of a
linearized system representing cyclo-stationary state of a model and implemented in a
biogeochemistry model, the Terrestrial Ecosystem Model (TEM).  With the new spin-up
algorithm, we showed that the model reached a steady state in less than 10 years of computing
time, while the original method requires more than 200 years on average of model run.  For the
test sites with five different plant function types, the new method saves over 90% of the original
spin-up time in site-level simulations. In North America simulations, average spin-up time
saving for all grid cells is 85% for either daily or monthly version of TEM.  The developed spin-
up method shall be used for future quantification of carbon dynamics at fine spatial and temporal
scales.

1. Introduction

Biogeochemistry models contain state variables representing various pools of carbon and nitrogen and a set of flux variables representing the element and material transfers between different state variables. Model spin-up is a step to get biogeochemistry models to a steady state for those state and flux variables (McGuire et al., 1992; King, 1995; Johns et al., 1997; Dickinson et al., 1998). Spin-up normally uses cyclic forcing data to force the model run, and reach a steady state, which will be used as initial conditions for model transient simulations. The steady state is reached when modeled state variables show a cyclic pattern or a constant and often requires a significant amount of computation time, which needs to be accelerated for regional and global simulations at fine spatial and temporal scales.

Spin-up is normally achieved by running model repeatedly using one or several decades of meteorological or climatic data, until a steady state is reached. The step could require model repeatedly run for more than 2000 annual cycles in some extreme cases. Specifically, the model will check the stability of the simulated carbon and nitrogen fluxes as well as state variables with specified threshold values. For instance, the model will check if the simulated annual net ecosystem production (NEP) is less than $1 \text{ g C m}^{-2} \text{ yr}^{-1}$ (McGuire et al. ,1992). Another method to reach a steady state is to obtain the analytical solutions (King et al, 1995; Comins, 1997), which might also take a significantly long time.

For different biogeochemistry models, spin-up could take hundreds and thousands of years to reach a stability, normally longer than the model projection period (Thornton et al., 2005). Therefore, a more efficient method to reach the steady state will speed up the entire model simulation. Recently, a semi-analytical method (Xia et al., 2012) has been adapted to a carbon-nitrogen coupled model to speed up the spin-up process. The idea is to get an analytical solution

very close to a steady condition, then start spin-up from the solution, which could significantly
reduce spin-up time. This technique did not reach a cyclic pattern for state and flux variables and
required an additional spin-up process to achieve the steady state. However, Lardy et al (2011)
and Martin et al (2007) have implemented their spin-up methods for a linear problem of soil
carbon dynamics including their seasonal cycles.
Here we developed a method to accelerate the spin-up process in a non-linear model.  We
tested the method for representative plant function types and the North America with both daily
and monthly versions of TEM (Zhuang et al., 2003). In addition, we compared the performance
of our algorithms with the semi-analytical version of Xia et al. (2012).  The new algorithms shall
help us conduct very high spatial and temporal resolution simulations with process-based
biogeochemistry models in the future.

2.  Method
2.1 TEM description
We used a process-based biogeochemistry model, the Terrestrial Ecosystem Model (TEM;
Zhuang et al. 2003) as testbed to demonstrate the performance of the new algorithms of spin-up.
TEM simulates the dynamics of ecosystem carbon and nitrogen fluxes and pools (McGuire et al.,
1992; Zhuang et al., 2010, 2003). It contains five state variables: carbon in living vegetation ($C_v$),
nitrogen in living vegetation ($N_v$), organic carbon in detritus and soils ($C_s$), organic nitrogen in
detritus and soils ($N_s$), and available inorganic soil nitrogen ($N_{av}$). Carbon and nitrogen
dynamics in TEM are governed by following equations:

$$\frac{dC_v}{dt} = GPP - R_A - L_C \dots\dots\dots\dots\dots\dots\dots\dots\dots\dots\dots\dots(1)$$

$$\frac{dN_v}{dt} = NUPTAKE - L_N \dots\dots\dots\dots\dots\dots\dots\dots\dots\dots(2)$$

$$\frac{dC_s}{dt} = L_c - R_H \dots\dots\dots\dots\dots\dots\dots\dots\dots\dots\dots\dots\dots(3)$$

$$\frac{dN_s}{dt} = L_N - NETNMIN \dots\dots\dots\dots\dots\dots\dots\dots\dots\dots(4)$$

$$\frac{dN_{av}}{dt} = NINPUT + NETNMIN - NLOST - NUPTAKE \dots..(5)$$

Where $GPP$ is gross primary production, $R_A$ is autotrophic respiration, $L_C$ is carbon in litterfall, $NUPTAKE$ is nitrogen uptake by vegetation, $L_N$ is nitrogen in litterfall, $R_H$ is heterotrophic respiration, $NETNMIN$ is net rate of mineralization of soil nitrogen, $NINPUT$ is nitrogen input from outside ecosystem, $NLOST$ is nitrogen loss from ecosystem. Key carbon fluxes are defined as:

$$GPP = C_{\max} f(PAR) f(PHENOLOGY) f(FOLIAGE) f(T) f(C_a, G_v) f(NA) f(FT) \dots\dots(6)$$

$$NPP = GPP - R_A \dots\dots\dots\dots\dots\dots\dots\dots\dots\dots\dots\dots\dots\dots\dots\dots\dots\dots\dots\dots\dots\dots\dots\dots\dots(7)$$

$$NEP = GPP - R_A - R_H \dots\dots\dots\dots\dots\dots\dots\dots\dots\dots\dots\dots\dots\dots\dots\dots\dots\dots\dots\dots\dots\dots\dots(8)$$

For detailed GPP definition, see Zhuang et al. (2003). NEP will be near zero when the ecosystem reaches a steady state. Therefore, the spin-up goal is to keep running the model driven with repeated climate forcing data until NEP is close to zero with a certain tolerance value (e.g., 0.1 g C m$^{-2}$ yr$^{-1}$).

2.2 Spin-up acceleration method

TEM can be re-formulated as:

$$\frac{d\vec{x}}{dt} = g\left(\vec{x},t\right) + \vec{h}\left(t\right)................................................................(9)$$

Where $\vec{x}$ is a vector of state variables (e.g., $C_V$); $\vec{h}\left(t\right)$ is the vector of carbon/nitrogen input
from the atmosphere (such as nitrogen input), independent on $\vec{x}$; $g\left(\vec{x},t\right)$ is the process rate
function of element pools (e.g., GPP).
By linearizing the model in term of pools, we could get:
$$g\left(\vec{x},t\right) = g\left(\vec{x}_0,t\right) + J\left(\vec{x} - \vec{x}_0\right)............................................(10)$$
Where $\vec{x}_0$ is initial pool sizes, J is the Jacobian matrix of the process rate:
$$J = \frac{dg}{dx} = \begin{bmatrix} \frac{\partial g_1}{\partial x_1} & . & . & . & \frac{\partial g_1}{\partial x_n} \\ . & & & & . \\ . & & & & . \\ . & & & & . \\ \frac{\partial g_n}{\partial x_1} & . & . & . & \frac{\partial g_n}{\partial x_n} \end{bmatrix} ....................................(11)$$
where g represents $g\left(\vec{x},t\right)$. $\vec{x}_n$ represents each of state variables in the TEM (e.g., $V_C$). The
numerical discretization of eq. (9) is:
$$x_{i,k} - x_{i,k-1} = \tau \cdot J_{k-\frac{1}{2}} \cdot x_{i,k-1} + \tau\left(g\left(x_0,k-1\right) - J_{k-\frac{1}{2}} \cdot x_{0,k-1} + h_{k-1}\right)........(12)$$

where $\tau$ is time step (month), $x_{i,k}$ is the pool $x_i$ size at time k, $J_{k-\frac{1}{2}}$ is a Jacobian matrix at
time step $k - \dfrac{1}{2}$. Here ½ refers to the half time step in the middle of a month, at which values
of J are calculated as the mean value at time steps k and k-1. $x_{i,0}$ refers to the initial pool $x_i$ size.
We introduce:
$$f_{k-1} = g\left(x_0, k-1\right) - J_{k-\frac{1}{2}} \cdot x_{0,k-1} + h_{k-1} \dots\dots\dots\dots\dots\dots\dots\dots\dots\dots\dots(13)$$
The eq. (12) can then be written as:
$$x_{i,k} - x_{i,k-1} = \tau \cdot J_{k-\frac{1}{2}} \cdot x_{i,k-1} + \tau \cdot f_{k-1} \dots\dots\dots\dots\dots\dots\dots\dots\dots\dots\dots(14)$$
Where $J_{k-\frac{1}{2}}$ is a Jacobian matrix at the time step $k - \dfrac{1}{2}$. After running a large number of
annual cycles, the model approaches a cyclo-stationary state, which can be expressed by
condition $x_{T+i} = x_i$, where T is the number of time steps in one cycle. For example, when spin
up is made at monthly time step using monthly climatology of temperature, precipitation and
other forcing data, T equals 12, and $\vec{x}_1$ is the size of carbon pools on January 1st, while $J^{1.5}$ is
the matrix of mean process rate constants for January.
By introducing:
$$A_k = \tau \cdot J_{k-\frac{1}{2}}, \; y_k = \tau f_{k-1}, \; B_k = I, \; C_k = I + A$$
where $I$ is an identity matrix.
Eq. (12) can be further written as:
$-C_k \cdot x_{i,k-1} + B_k \cdot x_{i,k} = y_k$ .......................................................(15)
The cyclic boundary condition is: $x_1 = x_{T+1}$
Then Eq. (13) will become:
$-C_1 \cdot x_{i,T} + B_1 \cdot x_{i,1} = y_1$ .......................................................(15a)
Thus eq. (15) and (15a) become a formulation of a linear problem with T unknown vectors
$\vec{x_T}$ , which can be solved using LU (lower and upper) decomposition or Gaussian elimination
(Martin et al., 2007). Xia et al (2012, see Eq. 4) and Kwon and Primeau (2006) also had linear
equations for a steady state, but conducted the model simulation at annual time step. Going for
annual average form reduces the size of the problem and prevents Xia et al (2012) from
obtaining the exact solution of the problem including seasonal cycle (see their Eq. 15, 15a).
While our new approach runs the model at monthly time step with the cyclic boundary
conditions for state variables x, it still targets a steady state for the ecosystem at annual time step
instead of monthly time step.
2.3 Numerical implementation
Eq. (15a) is explicitly expressed as:

$$
\begin{pmatrix}
B & 0 & 0 & ... & 0 & 0 & 0 & -C \\
-C & B & 0 & ... & 0 & 0 & 0 & 0 \\
0 & -C & B & ... & 0 & 0 & 0 & 0 \\
... & ... & ... & ... & ... & ... & .... & .... \\
0 & 0 & 0 & ... & -C & B & 0 & 0 \\
0 & 0 & 0 & ... & 0 & -C & B & 0 \\
0 & 0 & 0 & ... & 0 & 0 & -C & B
\end{pmatrix}
\times
\begin{pmatrix}
x_1 \\
x_2 \\
. \\
. \\
. \\
. \\
x_T
\end{pmatrix}
=
\begin{pmatrix}
y_1 \\
y_2 \\
. \\
. \\
. \\
. \\
y_T
\end{pmatrix}
\ldots\ldots\ldots\ldots.(16)
$$

Eq. (16) can be shown in form $Mx = Y$.
We apply the Gaussian elimination to upper block that reduces M to a lower triangular form
(Figure 1). The resulting matrix is lower diagonal:

$$M' = \begin{pmatrix} B' & 0 & 0 & 0 & 0 & 0 & 0 \\ -C & B & 0 & 0 & 0 & 0 & 0 \\ 0 & -C & B & 0 & 0 & 0 & 0 \\ \cdots & \cdots & \cdots & \cdots & \cdots & \cdots & \cdots \\ 0 & 0 & 0 & -C & B & 0 & 0 \\ 0 & 0 & 0 & 0 & -C & B & 0 \\ 0 & 0 & 0 & 0 & 0 & -C & B \end{pmatrix}$$

……………….(17)
The eq. (16) is thus reduced to form $M'x = Y'$, where $M'$ is lower diagonal, and solution of eq.
(15a and 16) will be readily obtained for x.
2.4 Algorithm implementation for TEM
In the original TEM, carbon fluxes can be defined as:

$$NPP = GPP - MR - GR \text{.............................................................(19)}$$

$$MR = V_C \cdot K_T \text{..................................................................(20)}$$

$$GR = \begin{cases} 0.25 \cdot (GPP - MR), & \text{if } GPP > MR \\ 0 & , \text{otherwise} \end{cases} \text{.......................................(21)}$$

Where net primary production (NPP) is defined as the difference of GPP and plant maintenance
respiration (MR) and growth respiration (GR). MR is assumed as a function of $C_V$ and
temperature ($K_T$). Here we revised MR calculation:
$$MR = \begin{cases} V_C \cdot K_T, & \text{if } GPP > V_C \cdot K_T \\ 0.75 \cdot V_C \cdot K_T + 0.25 \cdot GPP, & \text{otherwise} \end{cases} \text{..................(21)}$$
The net ecosystem production (NEP) is defined as the difference between NPP and
heterotrophic respiration ($R_H$).
The basic workflow to implement the method is: 1) linearizing TEM first to get a sparse
matrix with n-variable (for TEM, n=5) system; 2) performing Gaussian elimination for the linear
system; 3) solving the sparse matrix to acquire the state variable values (Figure 1). To adapt this
method to a daily version of TEM, we changed the cyclic condition T from 12 to 365. The other
steps are the same as monthly version. We tested the new method for carbon only version and
carbon-nitrogen coupled version of TEM for different plant functional types (PFTs) (Table 1).
Specifically, for the carbon only version, we only solved the differential equations that govern
the carbon dynamics, while for the carbon-nitrogen coupled version, we solved the differential
equations that govern both carbon and nitrogen dynamics in the system.  For the both versions,
the spin-up process strives to reach a steady state for carbon pools and fluxes.
3. Results and Discussion
At Harvard Forest site, the traditional spin-up method took 564 years to get the steady state
for both the carbon-only and coupled carbon–nitrogen simulations with annual NEP less than 0.1
g C $m^{-2}$ $yr^{-1}$ (Figure 2). In contrast, the improved method took 72 years for the carbon only and
122 for the coupled carbon–nitrogen simulations, respectively.  For carbon and nitrogen pools, it
took another 45 years (equivalent cyclic time) to reach a steady state with NEP less than 0.1 g C
$m^{-2}$ $yr^{-1}$.  In comparison with the traditional spin-up method (Zhuang et al., 2003), the new
method saved 65% of computational time to get the steady state in the carbon-only simulations
(Table 2). The differences in steady-state carbon pools between using the new method and
traditional spin-up methods were small (less than 0.85%).  Similarly, for the coupled carbon–
nitrogen simulations, the new method saves a similar amount of time to reach the steady state.
For all seven test sites, the original spin-up method in TEM takes 204-564 years (1.1-2.5
seconds of computing time) to reach the steady state at different sites. In contrast, our new
method only takes 0.3-0.6 seconds, while the semi-analytical method (Xia et al., 2012) will need
0.5-0.9 seconds to reach the steady state at different sites (Table 2). Compared to the original
spin-up method, the new method is not only faster, but also computationally stable.
The time of spin-up to reach a steady state of NEP varied for different PFT grids using the
original method (Figure 2). In general, to allow 98% grid cells reach their steady states of NEP,
it will take 250 annual model runs. While the new method will only need on average 0.6 seconds
(equivalent to 60-year annual model runs with the original method) (Figure 3). For regional tests
in North America, we found that the average saving time with the new method with monthly
TEM is 25%, 32%, and 22%, for Alaska, Canada, and the conterminous US, respectively.
To compare the performance of the new method with other existing methods, we adapted the
semi-analytical method (Xia et al., 2012) to TEM model. To do that, we first revised the TEM
model structure to:

$$\frac{dP(t)}{dt} = \varepsilon ACP(t) .............................. (22)$$


Where P(t) is a vector of pools in TEM (e.g., $C_V$ and $C_S$). $\varepsilon$ is a scalar. A is a pool transfer matrix
(in which $A_{ij}$ represents the fraction of carbon transfer from pool j to i). C is a diagonal matrix
with pool components (where diagonal components quantify the fraction of carbon left from the
state variables after each time step). With this method, we obtained an analytical solution for the
intermediate state. We then kept running TEM with the traditional spin-up process. Specifically,
we started TEM simulation to estimate the state variable values. Based on these values, the spin-
up runs were conducted to reach the final steady state. We found that the semi-analytical solution
is better than the original spin-up method, but slower than the new method proposed in this study
(Table 2).

194         The TEM model has a relatively small set of state variables for carbon and nitrogen.  The

version we used is TEM 5.0, which has only five state variables (Zhuang et al., 2003).  Thus, the
linearization process is relatively easy and the matrix size is relatively small, consequently, the
computing is not a burden.  To accelerate the spin-up for multiple soil carbon pool models with
relatively simple and linear decomposition processes, implementing our method shall be still
relatively easy, but will take a great amount of computing time to equilibrate. For models such as
CLM, multiple methods have been tested to accelerate their spin-up process (e.g., Fang et al.,
2015), the direct analytical solution method introduced in this study might be time-consuming to
achieve.
4. Summary

204         We developed a new method to speed up the spin-up process in process-based

biogeochemistry models. We found that the new method shortened 90% of the spin-up time
using the traditional method.  For regional simulations in North America, average spin-up time
saving is 85% for either daily or monthly version of TEM.  We consider our method is a general
approach to accelerate the spin-up process for process-based biogeochemistry models. As long as
the governing equations of the models can be formulated as the form in eq. (9), the algorithm
could be adopted accordingly. Our method will significantly help future carbon dynamics
quantification with biogeochemistry models at fine spatial and temporal scales.

Data availability: All data used in this study are available from the authors upon request.
Author contributions: Qianlai Zhuang and Shamil Maksyutov designed and supervised the
research. Yang Qu performed model simulations and data analysis. All authors contribute the
paper writing.
Competing interests: The authors declare that they have no conflict of interest.

Acknowledgements. This study is supported through projects funded to Qianlai Zhuang by
the Department of Energy (DESC0008092 and DE-SC0007007) and the NSF Division of
Information and Intelligent Systems (NSF-1028291). The super-computing resource is provided
by the Rosen Center for Advanced Computing at Purdue University.

Table 1. Test sites for new spin-up algorithms

| Site Name | Location | PFT | Reference |
|---|---|---|---|
| 1. Fort Peck | 48.3N, 105.1W | Grassland | Gilmanov et al. [2005] |
| 2. Bartlett Exp Forest | 44.1N, 71.3W | Deciduous broadleaf | Ollinger et al. [2005] |
| 3. UCI_1850 | 55.9N, 98.5W | Evergrenn needle-leaf | Goulden et al. [2006] |
| 4. Vaira Ranch | 38.4N, 121.0W | Grassland | Baldocchi et al. [2004] |
| 5. Missouri Ozark | 38.7N, 92.2 | Deciduous broadleaf | Gu et al. [2007, 2012] |
| 6. Niwot Ridge | 40.0N, 105.5W | Evergrenn needle-leaf | Turnipseed et al. [2003, 2004] |
| 7. Harvard Forest | 43.5N, 72.2W | Deciduous broadleaf | Van Gorsel et al. [2009] |




Table 2. Spin-up time comparison for different methods for seven study sites, seconds represent
real computation time, years refer to the spin-up annual cycles

| Site No. | Original Spin-up Year | Spin-up computation time (Seconds) | New method computation time (Seconds) | Semi-analytical method (equivalent annual cycles |
|---|---|---|---|---|
| 1 | 231 | 1.3 | 0.5 | 0.7s (+76) |
| 2 | 305 | 1.7 | 0.3 | 0.8s (+101) |
| 3 | 245 | 1.5 | 0.4 | 0.9s (+52) |
| 4 | 443 | 2.2 | 0.4 | 0.5s (+118) |
| 5 | 304 | 1.8 | 0.4 | 0.8s (+86) |
| 6 | 204 | 1.1 | 0.3 | 0.7s (+43) |
| 7 | 564 | 2.5 | 0.6 | 0.9s (+45) |



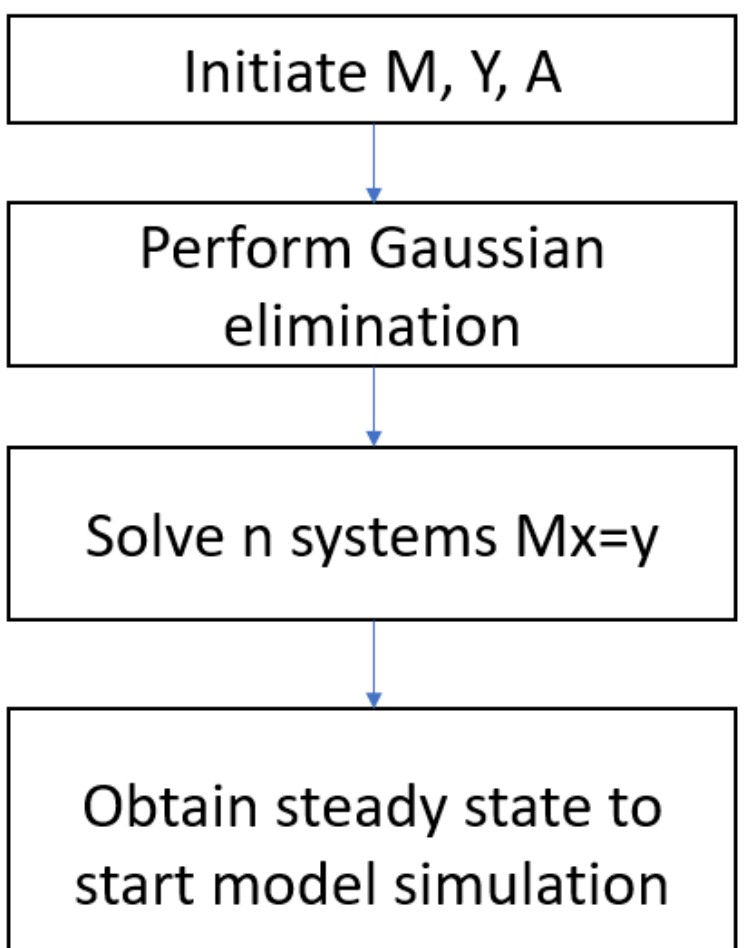


233       Fig. 1. Algorithms and procedures of the new spin-up method



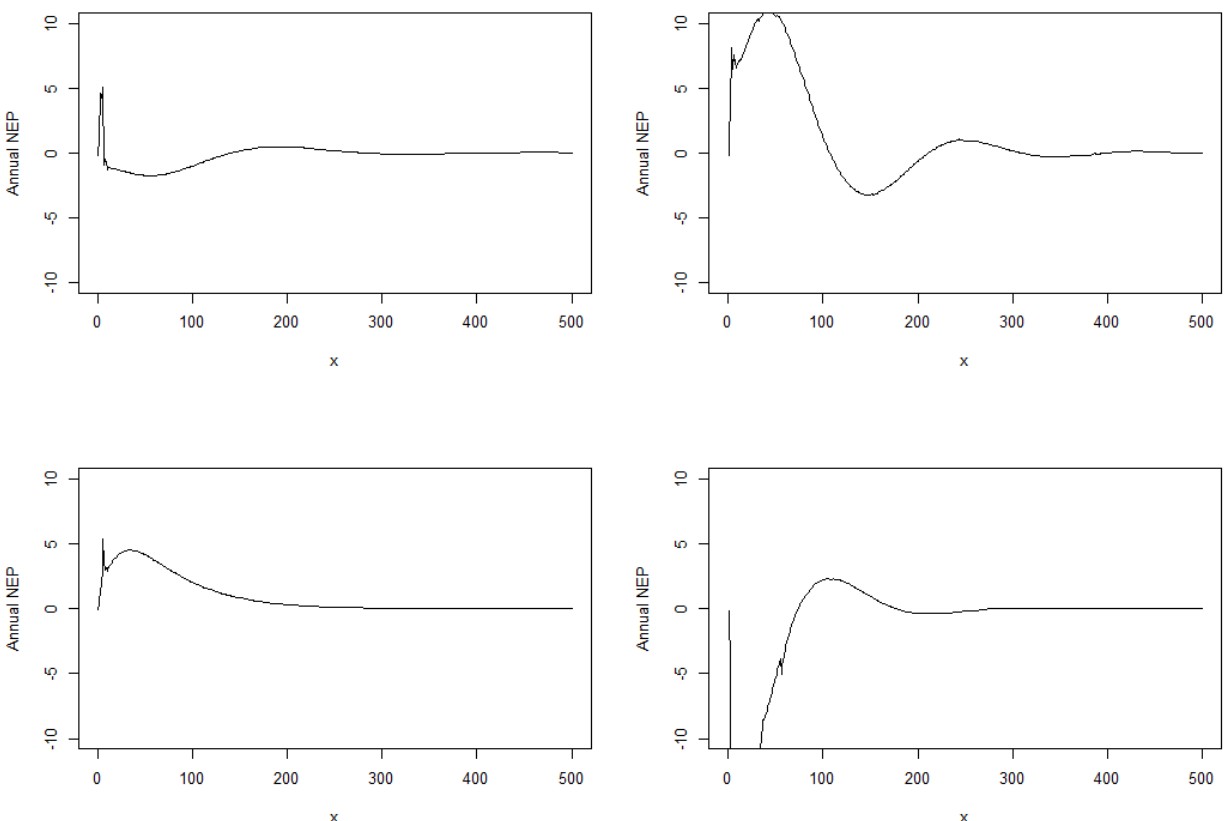


Fig. 2. The time for NEP (g C yr$^{-1}$m$^{-2}$) reached a steady state with the original spin-up method at

Harvard forest site. x represents model simulation years.




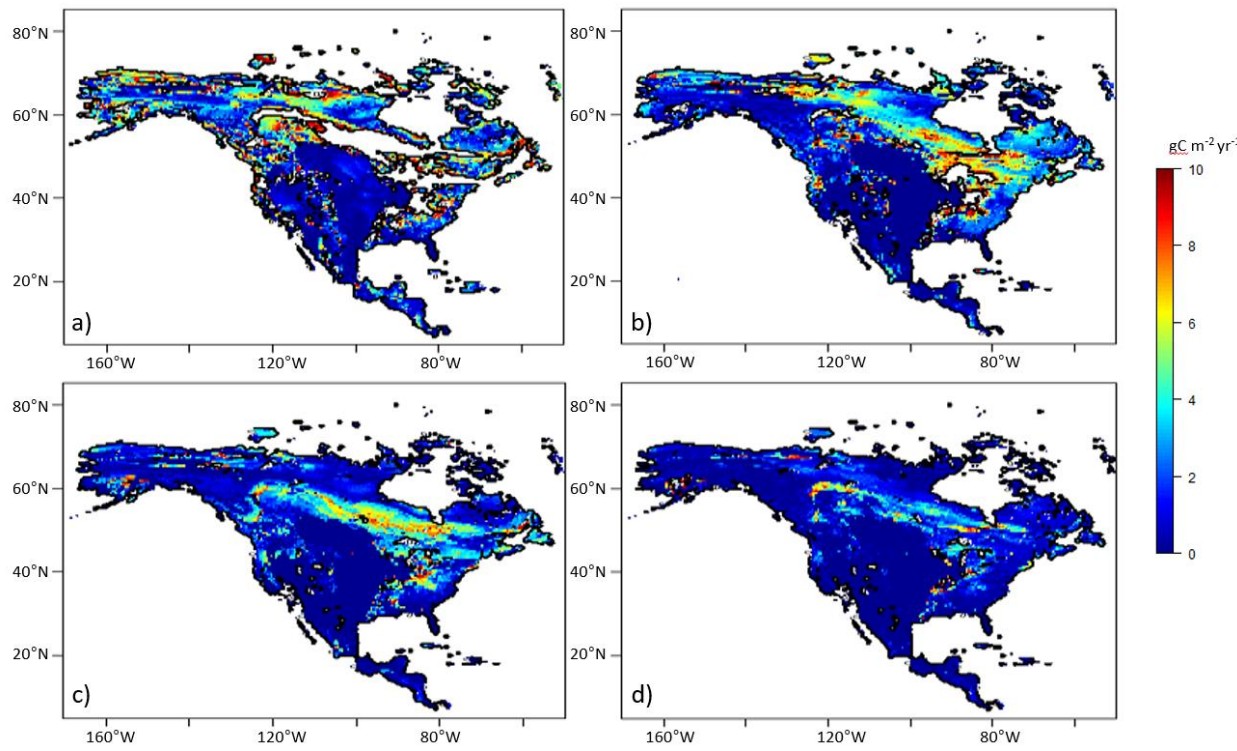


Fig. 3. Simulated NEP (g C m$^{-2}$ yr$^{-1}$) with the original spin-method after different spin-up years

of (a) 50, (b) 100, (c) 150, and (d) 200 years, respectively. After these spin-up years, 63%, 89%,

93%, and 98% grids will reach their steady states, respectively.

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
