# Peer review of "Technical Note: An efficient method for accelerating the spin-up process for process-based biogeochemistry models"

_Biogeosciences, 2018_

## Referee Comment (RC1) · Anonymous Referee #1 · 17 Mar 2018

This topic is of considerable interest, as spin-up provides the initial conditions for bio-geochemical model simulations but usually requires a significant amount of computation time. In the manuscript, authors want to propose a new method in order to accelerate spin-up processes at monthly and daily time steps, which modeled steady variables show a cyclic pattern. However, I am not completely convinced that the proposed method is efficient and reliable due to several reasons as follows:

1. Authors introduce many variables in Section 2.2, but most of them are not well illustrated. I feel confused about the difference between $k-1/2$ and $k-1$ in Eq. 12. If the spin-up is driven by monthly climatology, to my understanding, the Jk matrix should

[Figure]

depend on a constant matrix of transfer rate among pools and a matrix of pool size for each time step (k) in a specific year. It is vague and confusing that how to calculate the matrix of mean process rate constants (Line 104) for time step k (J(k-1/2)). Section 2.2 is the core of this new method, but authors simply list equations and do not explain how actually they have used it.

Additionally, in line 109, Eq. 12 can't be written as Eq. 15 when using yk=$\tau$f(k-1/2), but it is valid by using yk=$\tau$f(k-1). Authors should carefully check all the equations before submission.

2. In line 111, the cyclic boundary condition of this method is x1=x(T+1). As stated in the manuscript, when spin up is made at the monthly time step, T equals to 12 and x1 is the size of carbon pools in January. That means boundary condition is only applied to January carbon pools. This study mainly uses the Harvard Forest site (even though the authors listed seven sites in Table 1) which PFT is deciduous forests. The fluctuation of carbon pool/flux is the largest during the growing seasons. The method is not designed to reach a steady state for all state and flux variables during other months. As this method aims to derive a cyclic steady state, it is supposed to set a threshold or a boundary condition for each month/day or seasonal cycles, as well as for annual carbon balance (NEP).

3. Authors only present one table (Table 2) that contains the result of the proposed method, and other figures/tables are results from the original spin-up method. I didn't get the idea why to do this. In the Introduction (Line 42-43), it says that 'the model will check the stability of the simulated carbon and nitrogen fluxes as well as state variables with specified threshold values'. I didn't see much of these in Section 2 and 3.

Authors list seven sites to apply the new method, but I only see the results at Harvard Forest site, what about the results at other sites?

I would like to see more analyses and comparisons of flux trajectories and state variable trajectories for all carbon pools between original method and the new method

at site-level and pixel-level. The current results are not convincing to state the high efficiency of this new method.

Some specific comments:

Introduction: too many repetitive information in the first three paragraphs.

Line 42, is 2000 correct? In abstract, it is '200 years'. Need some references here.

Method:

Line 86, should Vc be written as Cv? Each variable name should be consistent throughout the manuscript.

Line 93, what does g(x0,k-1) represent? Please add explanations of each variable introduced in the equations.

Line 102, in boundary condition equation, T+i and i should be subscripted. Please check all the variable names.

Line 107-113, I suggest that all variables should be written as matrixes because authors introduce I as identify matrix in Line 107.

Line 115, please spell out LU.

Line 118-199, there is no Eq. 3a in the referred paper. Please check the citation.

Line 126-130, what are the matrixes with single quotes? What is D1 in line 127?

Line 127, it is unnecessary to be included in the manuscript.

Line 142, what is the exact value of n in TEM?

Results and Discussion:

Line 172-181, this paragraph would fit better in the Method.

All of the figures need more detailed captions.

References:

Line 252, loss of information.

---

## Referee Comment (RC2) · Anonymous Referee #2 · 16 Apr 2018

The spin-up issue is one of the technical bottlenecks for the modeling of biogeochemistry on the global scale. Using analytical approach to accelerate the spin-up of global biogeochemical models is promising, and this study provides a new approach with the TEM model. The method is novel and it can save ∼90% of the spin-up time. However, I have a few major concerns on the current version of the manuscript. A substantial revision is suggested before this work is considered for publication.

First, the section 2.2 is the core of this work, so the equations need more details and a double check. The equations (15) and (15a) are the fundamental equations in this method, but it is unclear how these two equations were derived from (12) and (13),

respectively. One major reason for the confusion is that the authors introduced the k-1/2 as a half time step. I encourage the authors to more carefully double check the equations.

Second, even the equations are all corrected, it is the authors' obligation to illustrate that how this approach could be adopted by other models. As we know, the structure of biogeochemistry in TEM used in this work is much simpler than those in many global land-surface models. For example, there are only two C pools and three N pools (see page 4), this makes the mathematical solution for the steady state much easier than those century-type models. I suggest the authors to add a section to discuss how their approach could be used in other models.

Third, the authors compared their new approach to the semi-analytical spin-up (SASU) method. The SASU method has shown that using analytical approach can dramatically save the spin-up time. However, many models still use the traditional methods of long-term iteration or some others, such as the accelerated decomposition (please see the technical note for the CLM4.5/5). The reason is that coding the analytical methods into the original model is time consuming. It would be great to see whether this method can save more time than those widely-used simple methods.

Forth, as shown in the Table 2, it seems the original TEM model reached the steady state very fast (∼200-500 years). This might be due to the short turnover times in the model (or the B components in the equation 16). Obviously the turnover of soil C is very slow at northern high latitudes. So it is not clear how the method will perform if the parameterizations for the soil module is realistic.

Below are some minor comments: 1. Pleask check eqn. (15) and (15a); 2. Page 4, Line 68-70: add a diagram to show the C-N structure of TEM would be helpful. 3. Page 6, Line 86: h is a vector of GPP? What is the difference between ""carbon input from the atmosphere" and "GPP"? 4. Page 9, Line 136: explain the "Vc"; 5. Fig. 1: please explain the contents in each box briefly in the figure leg-

end. 6. Fig. 3: unclear. The figures show the "spin-up time" or the steady-state C pool?

Please also note the supplement to this comment:
https://www.biogeosciences-discuss.net/bg-2018-98/bg-2018-98-RC2-supplement.pdf

---

## Author Comment (AC1) · 15 May 2018

Authors introduce many variables in Section 2.2, but most of them are not well illustrated. I feel confused about the difference between k-1/2and k-1 in Eq. 12. If the spin-up is driven by monthly climatology, to my understanding, the Jk matrix should depend on a constant matrix of transfer rate among pools and a matrix of pool size for each time step (k) in a specific year. It is vague and confusing that how to calculate the matrix of mean process rate constants (Line 104) for time step k (J(k-1/2)). Section2.2 is the core of this new method, but authors simply list equations and do not explainhow actually they have used it. Additionally, in line 109, Eq. 12 can't be written as Eq. 15

when using yk=τf(k-1/2), bu tit is valid by using yk=τf(k-1). Authors should carefully check all the equations before submission.

Response: In this revision, we clarified the definition of half-step Jacobian matrix and the way to compute it. Process rates depend on temperature and the process rate constants are time dependent, Index k-1/2 is introduced due to solving x(k)-x(k-1)= τ*(g(k-1/2)*x(k-1)+h(k-1/2)), which, as reviewer noted, would be commonly written as x(k)-x(k-1)= τ *(g(k-1)*x(k-1)+h(k-1)) in a purely explicit scheme. However, it can be shown that using process rates at midpoint (k-1/2) is no less accurate than in purely explicit form with (k-1), which is obvious for h(k-1/2). The term h(k-1/2) simply represents the value at the half-time step for function h. For Section 2.2, we revised it to show how each step is being done and how the next step is related to previous time step.

Line 115, please spell out LU Response: LU (Lower and Upper) decomposition refers to a matrix transformation to a Lower or Upper triangular form.

In line 111, the cyclic boundary condition of this method is x1=x(T+1). As stated in the manuscript, when spin up is made at the monthly time step, T equals to 12 and x1 is the size of carbon pools in January. That means boundary condition is only applied to January carbon pools. This study mainly uses the Harvard Forest site (even though the authors listed seven sites in Table 1) which PFT is deciduous forests. The fluctuation of carbon pool/flux is the largest during the growing seasons. The method is not designed to reach a steady state for all state and flux variables during other months. As this method aims to derive a cyclic steady state, it is supposed to set a threshold or a boundary condition for each month/day or seasonal cycles, as well as for annual carbon balance (NEP)

Response: Our model simulations are at a monthly step and we assume that, in a steady state, the pool sizes of x in December of the previous year shall equal the values in January of the next year. This treatment allow us reformulate the eq. 15 to

eq. 15a to efficiently obtain solutions for x. In addition, we still used the annual NEP to judge if the model has reached a steady state. In this revision, we revised the text to clearly state that our method is not to reach a monthly steady state for pools and fluxes, rather, we still target a steady state for the system at annual time step, in the end of the Section 2.2.

Authors only present one table (Table 2) that contains the result of the proposed method, and other figures/tables are results from the original spin-up method. I didn't get the idea why to do this. In the Introduction (Line 42-43), it says that 'the model will check the stability of the simulated carbon and nitrogen fluxes as well as state variables with specified threshold values'. I didn't see much of these in Section 2 and 3.Authors list seven sites to apply the new method, but I only see the results at Harvard Forest site, what about the results at other sites?

Response: In this revision, more details are added to the Result/Discussion sections. We also revise Table 2 to show the site-level results not only using the new method, but also other spin-up methods including a semi-analytical method and the original TEM spin-up method. To save space for a technical note paper, we intentionally only show the results at one site (Harvard forests) with Figure 2. We state that, similar results are also found for other sites in Table 1. In addition, we used Figure 3 to show regional results in North America to demonstrate the performance of the new method.

Please also note the supplement to this comment:
https://www.biogeosciences-discuss.net/bg-2018-98/bg-2018-98-AC1-supplement.zip

---

## Author Comment (AC2) · 15 May 2018

First, the section 2.2 is the core of this work, so the equations need more details and a double check. The equations (15) and (15a) are the fundamental equations in this method, but it is unclear how these two equations were derived from (12) and (13), respectively. One major reason for the confusion is that the authors introduced thek-1/2 as a half time step. I encourage the authors to more carefully double check the equations."

Response: In this revision, we clarified the definition of half-step Jacobian matrix J and the way to compute it. In our model, process rates depend on temperature and the

process rate constants are time dependent. Index k-1/2 is introduced due to solving x(k)-x(k-1)= $\tau$*(g(k-1/2)*x(k-1)+h(k-1/2)), which, as reviewer noted, would be commonly written as x(k)-x(k-1)= $\tau$*(g(k-1)*x(k-1)+h(k-1)) in a purely explicit scheme. However, it can be shown that using process rates at midpoint (k-1/2) is no less accurate than in purely explicit form with (k-1), which is obvious for h(k-1/2). The term h(k-1/2) simply represents the value at the half-time step for function h. For Section 2.2, we revised it to show how each step is being done and how the next step is related to previous time step.

Second, even the equations are all corrected, it is the authors' obligation to illustrate that how this approach could be adopted by other models. As we know, the structure of biogeochemistry in TEM used in this work is much simpler than those in many global land-surface models. For example, there are only two C pools and three N pools (see page 4), this makes the mathematical solution for the steady state much easier than those century-type models. I suggest the authors to add a section to discuss how their approach could be used in other models

Response: In this revision, we stated that "We consider our method is a general approach to accelerate the spin-up process for process-based biogeochemistry models. As long as the governing equations of the models can be formulated as the form in eq. (9), the algorithm could be adopted accordingly".

Third, the authors compared their new approach to the semi-analytical spin-up (SASU) method. The SASU method has shown that using analytical approach can dramatically save the spin-up time. However, many models still use the traditional methods of long-term iteration or some others, such as the accelerated decomposition (please see the technical note for the CLM4.5/5). The reason is that coding the analytical methods into the original model is time consuming. It would be great to see whether this method can save more time than those widely-used simple methods.

Response: We do consider our approach could be used for process-based models

with various structures. To accelerate the spin-up for multiple soil carbon pool models with relatively simple and linear decomposition processes, to implement our method is relatively easy, but will take a great amount of computing time to equilibrate. For models such as CLM, multiple methods have been tested to accelerate their spin-up process, but the direct analytical solution might be time-consuming to achieve. We added some discussion in the end of the Discussion section about this in this revision.

Forth, as shown in the Table 2, it seems the original TEM model reached the steady state very fast (âĹij200-500 years). This might be due to the short turnover times in the model (or the B components in the equation 16). Obviously the turnover of soil C is very slow at northern high latitudes. So it is not clear how the method will perform if the parameterizations for the soil module is realistic.

Response: Relatively quick spin up in TEM is due to the absence of slow soil carbon pool(s) as TEM has only one soil box (pool). Because soil carbon pools and litter pool are aggregated in TEM, the carbon turnover rates are dictated by fast turnover of litter pool, thus equilibration is faster than in CASA (Potter et al., 1993) or CENTURY (Parton et al., 1992) models. Accordingly, the benefit of our cyclo-stationary problem solver is less visible than in the case when it takes 2000 years or more to reach equilibrium. From eq. (17), we could see that our model performs stably for every grid in spite of the turnover rate, as the computation cost for LU decomposition is relatively stable. Added References: Potter, C. S., J. T. Randerson, C. B. Field, P. A. Matson, P. M. Vitousek, H. A. Mooney, and S. A. Klooster, 1993, Terrestrial ecosystem production: A process model based on global satellite and surface data, Global Biogeochemical Cycles, 7(4):811-841. Parton, W.J., B. McKeown, V. Kirchner, and D.S. Ojima. 1992. CENTURY Users Manual. Colorado State University, NREL Publication, Fort Collins, Colorado, USA.

Please also note the supplement to this comment:
https://www.biogeosciences-discuss.net/bg-2018-98/bg-2018-98-AC2-supplement.zip

---

## Author Response (AR1)

**Response letter**

Thanks for your constructive comments. Below we detail how we revised the manuscript following your suggestions.

(1) clarify the equations and method adopting the half time step approach

*Response: In this revision, we went through all equations to ensure they are correct. For instance, we change x to $\vec{x}$. We changed g(x) to $g\left(\vec{x},t\right)$ to make consistent in equations 9 and 10. In addition, we added the following sentence to clarify the half time step approach: "Here ½ refers to the half time step in the middle of a month, at which values of J are calculated as the mean value at time steps k and k-1. x0 refers to the initial pool size.".*

(2) underscore how the use of TEM may differ from a similar application with more complex models;

*Response: In this revision, we added some discussion on the implementation of this new spin-up method in TEM vs. in other models with more state variables, to underscore the differences implemented in TEM and other models: "The TEM model has a relatively small set of state variables for carbon and nitrogen. The version we used is TEM 5.0, which has only five state variables (Zhuang et al., 2003). Thus, the linearization process is relatively easy and the matrix size is relatively small, consequently, the computing is not a burden. To accelerate the spin-up for multiple soil carbon pool models with relatively simple and linear decomposition processes, implementing our method shall be still relatively easy, but will take a great amount of computing time to equilibrate. For models such as CLM, multiple methods have been tested to accelerate their spin-up process (e.g., Fang et al., 2015), the direct analytical solution method introduced in this study might be time-consuming to achieve.".*

(3) include a discussion, as in the response to R2, about the spin-up time achieved for TEM and how it relates to what would be achieved for applications to more complex models such as CASA and Century;

*Response: See our response (2) above. In addition, we added a few sentences to discuss how our new method could be applicable for other models in the Summary section: "We consider our method is a general approach to accelerate the spin-up process for process-based biogeochemistry models. As long as the governing equations of the models can be formulated as the form in eq. (9), the algorithm could be adopted accordingly.".*

(4) clarify the terminology for LU;

*Response: We added the full name for LU. The definition of LU is Lower and Upper operation.*

(5) re-iterate that the application seeks an annual steady state for a cyclo-stationary, monthly time step model rather than a monthly steady state

*Response: We made the statement clear, i.e., the spin-up method is targeting an annual steady state for a cyclo-stationary, rather than a monthly steady state, by adding "While our new approach runs the model at monthly time step with the cyclic boundary conditions for state variables x, it still targets a steady state for the ecosystem at annual time step instead of monthly time step.".*

(6) additional details on applications at other sites listed in Table 1, and the proposed revisions to Table 2

*Response: In this revision, we included spin-up performance for all 7 study sites with different spin-up methods as requested. These seven sites represent the key plant functional types in North America. The site information was documented in Table 1. Based on Table 2, we added a few sentences to describe the performance of the new method in comparison with other methods: "
[revised manuscript text omitted]
} \dfrac{\partial g_1}{\partial x_1} & . & . & . & \dfrac{\partial g_1}{\partial x_n} \\ . & & & & . \\ . & & & & . \\ . & & & & . \\ \dfrac{\partial g_n}{\partial x_1} & . & . & . & \dfrac{\partial g_n}{\partial x_n} \end{bmatrix} \quad\text{......................................(11)}$$

where g represents $g(\vec{x},t)$. $\vec{x}_n$ represents each of state variables in the TEM. The numerical discretization of equation (9) is:

$$x_{i,k} - x_{i,k-1} = \tau \cdot J_{k-\frac{1}{2}} \cdot x_{i,k-1} + \tau\left( g(x_0, k-1) - J_{k-\frac{1}{2}} \cdot x_{0,k-1} + h_{k-1} \right) \text{........(12)}$$

$$\sout{x_k - x_{k-1} = \tau \cdot J_{k-\frac{1}{2}} \cdot x_{k-1} + \tau\left( g(x_{0,k-1}) - J_{k-\frac{1}{2}} \cdot x_{0,k-1} + h_{k-1} \right) \text{........(12)}}$$

Where $\tau$ is time step (month), $x_k$ is the pool size at time k, $J_{k-\frac{1}{2}}$ is a Jacobian matrix at time step $k - \dfrac{1}{2}$. Here ½ refers the half timestep in the middle of a month, at which values of J are calculated as the mean value at time steps k and k-1. $x_{i,0}$ refers to the initial pool $x_i$ size.

We introduce:

$$f_{k-1} = g\left(x_{0,k-1}\right) - J \cdot x_{0,k-1} + h_{k-1} \quad\text{......................................(13)}$$

The eq. (12) can then be written as:

$$x_{i,k} - x_{i,k-1} = \tau \cdot J_{k-\frac{1}{2}} \cdot x_{i,k-1} + \tau \cdot f_{k-1} \quad\text{.....................................(14)}$$

Where $J_{k-\frac{1}{2}}$ is a Jacobian matrix at the time step $k - \dfrac{1}{2}$. After running a large number of annual cycles, model approaches a cyclo-stationary state, which can be expressed by condition $x_{T+i} = x_i$, where T is the number of time steps in one cycle. For example, when spin up is made at monthly time step using monthly climatology of temperature, precipitation and other forcing data, T equals 12, and $x^1$ is the size of carbon pools on January 1st, while $J^{1.5}$ is the matrix of mean process rate constants for January.

By introducing

$$A_k = \tau \cdot J_{k-\frac{1}{2}}, \; y_k = \tau f_{k-\frac{1}{2}}, \; B_k = I, \; C_k = I + A$$

where $I$ is an identity matrix.

Eq. (12) can be further written as:

$$-C_k \cdot x_{k-1} + B_k \cdot x_k = y_k \quad\text{.....................................................(15)}$$

The cyclic boundary condition is: $x_1 = x_{T+1}$

Then Eq. (13) will become:

$-C_1 \cdot x_T + B_1 \cdot x_1 = y_1$ ...........................................................(15a)

Thus eq. (15) and (15a) become a formulation of a linear problem with T unknown vectors $x_k$, which can be solved using LU (lower and upper) decomposition or Gaussian elimination (Martin et al., 2007). Xia et al (2012, see Eq. 4) and Kwon and Primeau (2006) also had linear equations for a steady state, but conduct the model simulation at annual time step. Going for annual average form reduces the size of the problem and prevents Xia et al (2012) from obtaining the exact solution of the problem including seasonal cycle (see their Eq. 15, 15a). While our new approach runs the model at monthly time step with the cyclic boundary conitions for state variables x, it still targets a steady state for the ecosystem at annual time step instead of monthly time step.

2.3 Numerical implementation

Eq. (15a) is explicitly expressed as:

$$
\begin{pmatrix}
B & 0 & 0 & \cdots & 0 & 0 & 0 & -C \\
-C & B & 0 & \cdots & 0 & 0 & 0 & 0 \\
0 & -C & B & \cdots & 0 & 0 & 0 & 0 \\
\cdots & \cdots & \cdots & \cdots & \cdots & \cdots & \cdots & \cdots \\
0 & 0 & 0 & \cdots & -C & B & 0 & 0 \\
0 & 0 & 0 & \cdots & 0 & -C & B & 0 \\
0 & 0 & 0 & \cdots & 0 & 0 & -C & B
\end{pmatrix}
\times
\begin{pmatrix}
x_1 \\ x_2 \\ . \\ . \\ . \\ . \\ x_T
\end{pmatrix}
=
\begin{pmatrix}
y_1 \\ y_2 \\ . \\ . \\ . \\ . \\ y_T
\end{pmatrix}
$$

$$
\begin{pmatrix}
B & 0 & 0 & 0 & 0 & 0 & -C \\
-C & B & 0 & 0 & 0 & 0 & 0 \\
0 & -C & B & 0 & 0 & 0 & 0 \\
\cdots & \cdots & \cdots & \cdots & \cdots & \cdots & \cdots \\
0 & 0 & 0 & -C & B & 0 & 0 \\
0 & 0 & 0 & 0 & -C & B & 0 \\
0 & 0 & 0 & 0 & 0 & -C & B
\end{pmatrix}
\times
\begin{pmatrix}
x^1 \\ x^2 \\ . \\ . \\ x^k \\ . \\ x^t
\end{pmatrix}
=
\begin{pmatrix}
y^1 \\ . \\ . \\ . \\ . \\ . \\ y^t
\end{pmatrix}
,
$$
.................(16)

Eq. (16) can be shown in form $Mx = Y$.

We apply the Gaussian elimination to upper block that reduces M to a lower triangular form (Figure 1). The elimination process is applied from right to left in the top row of M involving 2x2 blocks of matrices $\underline{B_k, C_k, D \text{ and } D^1}$.

$$\begin{pmatrix} D^1 & D \\ -C_k & B_k \end{pmatrix} \begin{pmatrix} y_1 \\ y_k \end{pmatrix}$$ …………………………………….(17)

The resulting matrix is lower diagonal:

$$M' = \begin{pmatrix} B' & 0 & 0 & 0 & 0 & 0 & 0 \\ -C & B & 0 & 0 & 0 & 0 & 0 \\ 0 & -C & B & 0 & 0 & 0 & 0 \\ \cdots & \cdots & \cdots & \cdots & \cdots & \cdots & \cdots \\ 0 & 0 & 0 & -C & B & 0 & 0 \\ 0 & 0 & 0 & 0 & -C & B & 0 \\ 0 & 0 & 0 & 0 & 0 & -C & B \end{pmatrix}$$ ………………(18)

The eq. (16) is thus reduced to form $M'x = Y'$, where $M'$ is lower diagonal, and solution of eq. (15a and 16) will be readily obtained for x.

2.4 Algorithm implementation for TEM

In the original TEM, carbon fluxes can be defined as:

$$NPP = GPP - MR - GR \text{.................................................................(19)}$$
$$MR = V_C \cdot K_T \text{.......................................................................(20)}$$
$$GR = \begin{cases} 0.25 \cdot (GPP - MR), & \text{if } GPP > MR \\ 0 & \text{, otherwise} \end{cases} \text{.....................................(21)}$$

Where net primary production (NPP) is defined as the difference of GPP and plant maintenance respiration (MR) and growth respiration (GR). MR is assumed as a function of $C_V$ and temperature ($K_T$). Here we revised MR calculation:

$$MR = \begin{cases} V_C \cdot K_T, & if\ GPP > V_C \cdot K_T \\ 0.75 \cdot V_C \cdot K_T + 0.25 \cdot GPP, & otherwise \end{cases} \quad \text{..................(21)}$$

The net ecosystem production (NEP) is defined as the difference between NPP and heterotrophic respiration ($R_H$).

The basic workflow to implement the method is: 1) linearizing TEM first to get a sparse matrix with n-variable (for TEM n=5) system; 2) performing Gaussian elimination for the linear system; 3) solving the sparse matrix to acquire the state variable values (Figure 1). To adapt this method to a daily version of TEM, we changed the cyclic condition T from 12 to 365. The other steps are the same as monthly version. We tested the new method for carbon only version and carbon-nitrogen coupled version of TEM for different plant functional types (PFTs) (Table 1). Specifically, for the carbon only version, we only solved the differential equations that govern the carbon dynamics, while for the carbon-nitrogen coupled version, we solved the differential equations that govern both carbon and nitrogen dynamics in the system. For the both versions, the spin-up process strives to reach a steady state for carbon pools and fluxes.

3. Results and Discussion

At Harvard Forest site, the traditional spin-up method took 564 years to get the steady state for both the carbon-only and coupled carbon–nitrogen simulations with annual NEP less than 0.1 g C m$^{-2}$ yr$^{-1}$ (Figure 2). In contrast, the improved method took 72 years for the carbon only and 122 for the coupled carbon–nitrogen simulations, respectively. For carbon and nitrogen pools, it took another 45 years (equivalent cyclic time) to reach a steady state with NEP less than 0.1 g C m$^{-2}$ yr$^{-1}$. In comparison with the traditional spin-up method (Zhuang et al., 2003), the new method saved 65% of computational time to get the steady state in the carbon-only simulations (Table 2). The differences in steady-state carbon pools between using the new method and traditional spin-up methods were small (less than 0.85%). Similarly, for the coupled carbon–

nitrogen simulations, the new method saves a similar amount of time to reach the steady state. The new method performs similarly for the rest of six sites.

For all seven test sites, the original spin-up method in TEM takes 204-564 years (1.1-2.5 seconds of computing time) to reach the steady state at different sites. In contrast, our new method only takes 0.3-0.6 seconds, while the semi-analytical method (Xia et al., 2012) will need 0.5-0.9 seconds to reach the steady state at different sites (Table 2). Compared to the original spin-up method, the new method is not only faster, but also computationally stable.

For all seven test sites, it takes on average 0.6 seconds using the new method to reach a steady state. Compared to the original spin-up method, the new method is not only faster, but also computationally stable.

[revised manuscript text omitted]

```
┌─────────────────────────┐
│     Initiate M, Y, A    │
└─────────────────────────┘
              │
              ▼
┌─────────────────────────┐
│    Perform Gaussian     │
│       elimination       │
└─────────────────────────┘
              │
              ▼
┌─────────────────────────┐
│   Solve n systems Mx=y  │
└─────────────────────────┘
              │
              ▼
┌─────────────────────────┐
│   Obtain steady state to│
│  start model simulation │
└─────────────────────────┘
```

[Figure]

Initiate M, y, A

Solve n systems Mx=y

Perform Gaussian elimination in
(D', D / C, B), zeroing D

Solve Sparse:
$B_kx_k=y_k+C_kx_{k-1}$
For k=2:t

Acquire initial steady state to start
model simulation

Fig. 1. Algorithms and procedures of the new spin-up method

[revised manuscript text omitted]